# A Comparison of the Effects of Milk, Yogurt, and Cheese on Insulin Sensitivity, Hepatic Steatosis, and Gut Microbiota in Diet-Induced Obese Male Mice

**DOI:** 10.3390/ijms26115026

**Published:** 2025-05-23

**Authors:** Emad Yuzbashian, Dineli N. Fernando, René L. Jacobs, Till-Robin Lesker, Till Strowig, Siegfried Ussar, Catherine B. Chan

**Affiliations:** 1Department of Agricultural, Food and Nutritional Science, University of Alberta, Edmonton, AB T6G 2P5, Canada; yuzbashi@ualberta.ca (E.Y.); rjacobs@ualberta.ca (R.L.J.); 2Department of Cell Biology, University of Alberta, Edmonton, AB T6G 2H7, Canada; wfernand@ualberta.ca; 3Department of Microbial Immune Regulation, Helmholtz Center for Infection Research, 38124 Braunschweig, Germany; till-robin.lesker@helmholtz-hzi.de (T.-R.L.); till.strowig@helmholtz-hzi.de (T.S.); 4Helmholtz Center for Individualized Infection Medicine, 30625 Hannover, Germany; 5Research Unit Adipocytes and Metabolism, Helmholtz Diabetes Center, Helmholtz Zentrum München, Research Center for Environmental Health GmbH, 85764 Neuherberg, Germany; siegfried.ussar@helmholtz-munich.de; 6German Center for Diabetes Research (DZD), 85764 Neuherberg, Germany; 7Department of Physiology, University of Alberta, Edmonton, AB T6G 2H7, Canada

**Keywords:** dairy, microbiome, lipidomics, diabetes, obesity

## Abstract

The effects of low-fat dairy products on insulin resistance (IR), hepatic steatosis, and gut microbiota composition in high-fat diet (HFD)-fed obese mice were examined. C57BL/6 male mice (n = 16/group) were fed a high-fat diet (HFD, 45% fat) or HFD supplemented with either fat-free milk (MILK), fat-free yogurt (YOG), or reduced-fat (19% milk fat) cheddar cheese (CHE) at 10% of the total energy intake for 8 weeks. Body weight, fat mass, liver lipids, and metabolic enzymes were evaluated. Compared with HFD, MILK reduced homeostatic assessment of insulin resistance along with increased hepatic insulin signaling and decreased hepatic gluconeogenic enzymes. YOG and MILK decreased hepatic triacylglycerol content and lipid droplet size, while CHE had no effect. In the liver, MILK and YOG downregulated de novo lipogenesis enzymes. In MILK, fat oxidation capacity was elevated. Compared with HFD, liver lipidomic analysis in MILK and YOG revealed unique profiles of decreased proinflammatory lipid species, including ceramides. Dairy feeding elicited an increase in beneficial bacteria, such as *Streptococcus* in YOG and *Anaero-tignum* in MILK, as shown by 16S rRNA sequencing of gut microbiota. In conclusion, the ability of milk and yogurt to reduce hepatic steatosis in HFD mice may be explained, at least in part, by the regulation of the gut microbiome and liver lipidome.

## 1. Introduction

Metabolic dysfunction-associated steatotic liver disease (MASLD) is characterized by the accumulation of triacylglycerol (TG) in hepatocytes in the absence of excess alcohol intake. The worldwide prevalence of MASLD is currently 32% among adults, and continues to rise [1,2]. Because of its strong association with obesity and insulin resistance (IR), MASLD is considered the hepatic manifestation of obesity-induced metabolic dysfunction [3]. Its increased prevalence is associated with the consumption of Western-style diets, which are characterized by processed foods, red meat, and refined grains [4]. Growing evidence indicates that Western-style diets disrupt the gut microbiota composition and function (dysbiosis), which is implicated in the development of MASLD and the progression of metabolic impairment [5]. Thus, diet modification is among the leading strategies for preventing and treating MASLD and associated risk factors [6]. The lack of approved pharmaceutical treatments for MASLD until recently [7] and the importance of optimizing diet in its prevention provide impetus for evidence-based public health initiatives focused on specific food groups that contribute to metabolic health.

Dairy foods are nutrient-dense, supplying high-quality protein and beneficial micronutrients and are foundational in many healthy dietary patterns [8]. A recent meta-analysis of observational studies indicates that higher total dairy consumption is associated with a 10% lower risk of liver steatosis [9]. A large prospective cohort study reports that consuming low-fat dairy is associated with a lower risk of incident liver steatosis than high-fat dairy (−22% vs. −3%) [10]. However, dairy products, including milk, yogurt, and cheese, differ in their nutritional composition and physicochemical properties (matrices) due to specific fermentation and processing methods. The dairy matrix influences digestion, bioavailability, and metabolic effects [11]. Therefore, the impact of consuming individual dairy products on metabolic health may vary.

Liquid milk is beneficially associated with metabolic health and the prevention of hepatic steatosis [12,13]. Preclinical studies indicate that skimmed milk powder intake yields improvement in metabolic parameters [14,15]. Because of the link between gut dysbiosis and the development of MASLD [16], fermented milk products containing probiotics are assumed to add health-related value [17]. Yogurt, a semi-solid probiotic food produced through milk fermentation using certain lactic acid bacteria such as *Streptococcus thermophilus* and *Lactobacillus delbrueckii* subsp-bulgaricus, contains these beneficial bacteria and their metabolites [18]. A solid milk derivative, cheese, is also made by fermenting milk with lactic acid bacteria, specifically *Lactococcus lactis* and *L. cremoris*, with an additional ripening step [19]. In addition to being nutrient-rich, fermented dairy products may prevent MASLD development because they contain probiotics that can modify the gut microbiome structure or function, in addition to fermentation-derived bioactive compounds [17,20]. Recent clinical trials report that non-fermented milk products and yogurt have similar abilities to reduce liver steatosis but impact metabolic risk markers differently, suggesting that distinct mechanisms might be involved [21].

The mechanisms by which dairy products contribute to metabolic benefits, particularly in their natural matrix form, remain to be defined. Additionally, it is unclear whether these effects are related to changes in gut microbiota composition. Thus, this study aimed to compare the effect of fat-free milk, plain fat-free yogurt, and reduced-fat cheddar cheese consumption on obesity phenotype markers, IR, and hepatic steatosis in mice fed a high-fat diet (HFD). Additionally, we sought to understand how the gut microbiota and liver metabolic pathways might contribute to these effects. We hypothesized that both milk and fermented dairy product consumption would ameliorate the detrimental metabolic effects of a HFD, leading to protection against hepatic steatosis through both shared and unique mechanisms. Moreover, we anticipated that consuming fermented dairy products would significantly improve metabolic risk indicators and reduce liver fat content compared with consuming milk.

## 2. Results

### 2.1. HFD-Induced Obesity and Development of MASLD

Compared with a low-fat diet (LFD), 8 weeks of a high-fat diet (HFD) fed to C57BL/6 male mice caused greater energy intake and body weight gain, a lower lean-to-fat mass ratio, dysglycemia (higher fasting blood glucose (FBG) and homeostatic model assessment of insulin resistance (HOMA-IR)), and hyperlipidemia (elevated serum TG and total cholesterol (TC) after fasting-refeeding) (Appendix A). In addition, compared with LFD, HFD had significantly greater liver TG and TC accumulation in both fasted and refed states (Appendix A). This was confirmed by histology, which demonstrated larger, more plentiful lipid droplets in HFD mice (Appendix A), thereby validating the successful establishment of a MASLD model.

### 2.2. Effect of Dairy Product Consumption on Obesity Phenotypes and Glucose and Lipid Parameters

While all mice started the study with similar body weights, by day 56 (D56) mice in the MILK, YOG, and CHE groups had lower body weight, weight gain, and fat mass compared with those fed HFD (Table 1). MILK and YOG had significantly greater lean-to-fat mass ratios than HFD. Although the average daily energy intake did not differ among the groups, MILK, YOG, and CHE had greater weight-adjusted energy expenditure compared with HFD. The FBG at D0 was similar across all groups but was significantly lower in the MILK, YOG, and CHE groups than in the HFD group at D56. Compared to HFD, MILK had significantly lower serum fasting insulin and HOMA-IR, indicating higher insulin sensitivity. There was a trend toward lower fasting insulin and HOMA-IR in the YOG and CHE groups compared with HFD (*p* < 0.1). During fasting, serum TG was not different between groups, but refed mice in MILK and YOG had lower serum TG than HFD. No differences in serum TC were observed.

### 2.3. Effect of Dairy Product Consumption on Accumulation of Fat and Lipid Metabolism Pathways in the Liver

Microscopic examination of the liver histology samples (Figure 1A) revealed smaller lipid droplets in all dairy-fed mice compared with the HFD group. Compared with HFD in both fasted (Figure 1B) and refed states (Appendix A), MILK had the most notable reduction in hepatic lipid droplet size, followed by YOG and CHE. The liver TG content in MILK and YOG was about one-third that in HFD in the fasting state (Figure 1C), with a similar trend toward decreased liver TG content in MILK and YOG in the refed state (*p* < 0.1) (Appendix A). No differences were found in hepatic TC content (Figure 1D) between the groups.

Immunoblotting was conducted on livers collected in both the refed and fasted states to assess the abundance of regulatory proteins involved in hepatic lipid metabolism and to explore potential mechanisms for reduced lipid droplet size and hepatic TG in dairy-fed mice. The most marked results were observed in the fasting state (Figure 1E–K). Compared with HFD, phosphorylated AMP-activated protein kinase-α (pAMPK)/AMPK was greater in MILK and YOG (Figure 1E). Greater sirtuin 1 (SIRT1) and peroxisome proliferator-activated receptor-α (PPAR-α) proteins (Figure 1F,G) were also detected when comparing the MILK and HFD groups. Greater peroxisome proliferator-activated receptor gamma coactivator-1α (PGC-1α) (Figure 1H) was observed in the YOG group compared with the HFD control group. Regarding lipid oxidation pathways, long-chain acyl-CoA synthetase-1 (ACSL-1) was similar among groups (Figure 1I), but carnitine palmitoyltransferase-1α (CPT-1α) was greater by almost 2-fold in all three dairy intervention groups compared with the HFD group (Figure 1J). In the refed state, YOG increased pAMPK/AMPK by 1.8-fold (Appendix A), while MILK elicited an increase in adipose triglyceride lipase (ATGL), which catalyzes lipolysis (Appendix A).

### 2.4. Effect of Dairy Product Consumption on Liver Lipogenesis, Lipid Import and Export Pathways

PPAR-γ promotes fatty acid storage in the liver. All groups had similar abundance of PPAR-γ in both the fasted (Figure 2A) and refed (Appendix A) states. Compared with HFD in the fasting state, MILK, YOG, and CHE groups exhibited greater phosphorylation of acetyl-CoA carboxylase (pACC/ACC), a rate-limiting enzyme whose phosphorylation suppresses de novo lipogenesis (DNL) (Figure 2B). Also in the fasting state, no significant differences among the groups were observed in fatty acid synthase (FAS) (Figure 2C), phosphorylated ATP-citrate lyase (pACL)/ACL (Figure 2D), and acetyl-CoA synthetase (AceCS) (Figure 2E). In the refed state, there were no significant differences in pACC/ACC, FAS, pACL/ACL, and AceCS among groups (Appendix A). The abundance of microsomal triglyceride transfer protein (MTP; lipid export) and cluster of differentiation 36 (CD36; fatty acid uptake) did not change in response to dairy intervention in either fasted (Figure 2F,G) or refed states (Appendix A). Relative to the HFD group, all dairy groups had similar fibroblast growth factor 21 (FGF21) levels in the fasted (Figure 2H) and refed states (Appendix A). No differences in OXPHOS complex abundance were detected (Figure 2I and Appendix A).

### 2.5. Effect of Dairy Product Consumption on Insulin-Regulated Liver Enzymes

Compared with HFD, MILK exhibited greater phosphorylation of AKT (pAKT/AKT), a marker of hepatic insulin action, in the refed state with a similar trend in the fasted state (Figure 3A). Consistent with enhanced insulin sensitivity, further investigation of enzymes controlling hepatic glucose production revealed decreased abundance of essential gluconeogenic enzymes, including phosphoenolpyruvate carboxykinase (PEPCK) among MILK, YOG, and CHE in the refed state (Figure 3B) and glucose 6-phosphatase (G6P) in YOG compared with HFD (Figure 3C). In the fasting state, PEPCK was lower in MILK than in HFD (Figure 3B).

### 2.6. Effect of Dairy Product Consumption on the Liver Lipidome

Quantitative analysis of liver lipid metabolites was performed to compare overall metabolic changes in mice fed an HFD alone or with dairy supplementation. Unsupervised principal component analysis (PCA) (Appendix A) and partial least squares discriminant analysis (PLS-DA) (Figure 4A) did not yield distinct clusters of liver lipid species among groups (leave-one-out cross-validation (LOOCV) R^2^ = 60%, 1000 permutations *p*-value = 0.376). However, supervised (s)PLS-DA models, which provide better discrimination by considering 100 of the most discriminative features, demonstrated clustering by dairy intervention with a clear separation of the MILK group from the YOG, CHE, and HFD groups (Figure 4B) (LOOCV R^2^ = 87.5%). Variable importance in projection (VIP) scores were used to identify the lipid molecules most responsible for the clustering (Figure 4C) and yielded phosphatidylserine (PS) 38:1, DG O-42:3, hexosylceramide (HexCer):40:2;O2, prenol (PR); 23:3;O2, sulfatide (SHexCer) 38:6;O, phosphatidylcholine (PC) -34:4, and TG 41:1′O. A heatmap (Figure 4D) was generated from group averages with clustering by Ward linkage of the top 25 highly significant differences in lipids identified in the VIP, revealing discrepancies between groups. The classification of significantly altered lipids is provided in Appendix A. Unique lipidomic profiles were demonstrated through these analyses, with MILK and YOG illustrating a distinct pattern from MILK and CHE, which were similar.

Compared with HFD mice, in the MILK group, 252 hepatic lipid species were significantly greater and 246 lower (total 498) (Figure 4E), while 397 lipids were greater and 267 lipids were lower (total 664) in YOG vs. HFD (Figure 4F). In CHE compared with HFD, 253 lipids were greater and 175 lipids were lower (total 428) (Figure 4G). Compared with HFD, 80 lipids were consistently greater in all three dairy intervention groups (Figure 4H, including 13 TG, 8 DG, 10 Cer, 6 HexCer, 4 PI, 4 PS, 6 PE, 5 phosphatidylglycerol (PG), and various other lipids. Conversely, 34 lipids were consistently lower (Figure 4I): 6 PG, 4 HexCer, 3 DG, 5 TG, among others (Appendix A). Several lipid species were consistently upregulated in the MILK and YOG groups but not in the CHE group (Appendix A), for instance, DG O-39:0;O2 and DG O-61:5, Cer 44:6;O3 and Cer 38:5;O2, and TG 74:3;O2 and TG 16:0_22:0_22:4. Additionally, distinct unique lipid signatures in the MILK and YOG were found, with several differentially upregulated lipid species that may be related to beneficial effects on liver fat accumulation (Appendix A).

Hepatic lipid metabolism altered by dairy interventions was identified by lipid ontology (LION enrichment analysis). Compared with HFD, downregulated species in the MILK group were linked to diradylglycerols (GL02), diacylglycerols (GL0201), glycerolipids (GL), and headgroups with a neutral charge. In the YOG group, LION analysis identified a significant reduction of pathways involving fatty acids with three double bonds, sphingolipids (SP), C20:3, steryl esters (ST0102), diacylglycerols (GL0201), diacylglycerophosphocholines (GP1010), fatty acids with 20 carbons, hexosylceramides, and ceramide phosphocholines (sphingomyelins) (SP0301) (Figure 4J). YOG also had upregulated pathways linked to lipid headgroups with negative charge, C12:0, glycerophospholipids (GP), positive intrinsic curvature, triradylglycerols (GL03), glycerophosphates (GP10), and triacylglycerols (GL0301) (Figure 4K). CHE elicited no significant upregulation or downregulation in specific lipid pathways compared with HFD (Appendix A).

### 2.7. Effect of Dairy Product Consumption on the Gut Microbiome

The gut microbiome composition was compared to assess its potential contribution to the metabolic effects of dairy intake. α-Diversity analysis was assessed. The number of observed operational taxonomic units (OTUs), which indicate within-group microbiota richness, was not different between the groups (Figure 5A). However, bacterial richness and evenness in the YOG group were significantly greater compared with the HFD and CHE groups, as evidenced by the Shannon (Figure 5B) and Simpson indices (Figure 5C). 

A β-diversity index was used to estimate the differences in microbiota composition between the four groups. Bray–Curtis dissimilarity analysis indicated a difference between groups (*p* = 0.012), and post hoc testing illustrated pairwise differences between MILK–HFD and MILK–YOG groups (Figure 5D). The same pattern was observed when β-diversity was calculated based on unweighted UniFrac distance (Figure 5E, *p* = 0.001) and weighted UniFrac distance (Figure 5F, *p* = 0.012), indicating the bacterial microbiota of MILK clustered apart from that of HFD and YOG. Compared with HFD, DESeq log 2-fold changes analysis (Figure 5G) found that MILK supplementation enriched the abundance of genera *Anaerotignum_189125* and *oscillospiraceae_88309* while depleting *Avispirillum*, *Longicatena*, and *Turicibacter* genera. YOG supplementation significantly increased the abundance of *Streptococcus* and *Clostridium*, and decreased *Evtepia* and *Lachnospiraceae* abundance compared with HFD, while CHE treatment enriched *Sporofaciens* and *Streptococcus* and depleted *Limivicinus* abundance.

## 3. Discussion

The present study compared the effect of consuming milk, yogurt, and cheese in amounts equivalent to two servings per day by humans (about 10% of total energy intake) on outcomes related to hepatic steatosis. Dairy products mitigated the development of obesity, IR, and hepatic steatosis in HFD-induced obese mice to varying degrees. Specifically, mice fed MILK, YOG, and CHE had lower weight gain and greater energy expenditure, as reported in more detail in our previous publication [22]. Contrary to our hypotheses regarding the superiority of fermented compared with non-fermented dairy products, MILK significantly enhanced insulin sensitivity (reduced HOMA-IR), and both MILK and YOG reduced hepatic TG. The beneficial impact of MILK and YOG may have been partly driven by changes in gut microbiota composition. However, the benefits of MILK and YOG were associated with distinct modifications of the hepatic lipidome and gut microbiome compared with HFD, suggesting unique mechanisms. These data highlight the significant influence of dairy products on metabolic pathways, based on their unique matrices.

Recent randomized controlled trials provide evidence that milk [21,23] and plain yogurt [24], as part of a habitual diet, improve IR and liver fat in humans. However, the mechanisms of action of milk and yogurt, as well as the hepatic contributions to this improvement, remain unclear. HFD-induced obesity and IR impair the activation of AKT, leading to dysregulated gluconeogenesis and glycogenolysis. Insulin fails to suppress the expression of *Pepck* and *G6pc1* mRNA during the refeeding period, leading to greater glucose release into circulation, resulting in hyperglycemia [25,26]. MILK and YOG reduced FBG, likely via reduced hepatic gluconeogenesis as evidenced by lower PEPCK compared with HFD, and consistent with improved hepatic insulin signaling, as supported by increased AKT phosphorylation. Although few studies report tissue-specific mechanisms following milk consumption in animals, the beneficial effects of milk components, such as whey protein [27] and polar lipids [28], in mitigating the detrimental effect of HFD on glucose homeostasis have been observed [29]. Whey protein supplementation effectively normalizes *Pepck* and *G6pc1* in the livers of mice fed an HFD compared with an LFD, the healthy control group [30]. Also, in HFD-induced IR mice, yogurt consumption reduces HOMA-IR [31,32] and improves insulin sensitivity in the liver by augmenting phosphatidylinositol-3-phosphate and AKT signaling while suppressing PEPCK [31]. Here, the lower PEPCK and G6P in the refed state observed in the YOG group did not result in improved HOMA-IR, which is a fasting indicator of IR. For this reason, the interpretation of the molecular data obtained in this study is limited by the lack of functional assessment of insulin sensitivity beyond HOMA-IR.

In humans, prospective cohort studies [12,33] and meta-analyses [9,34] suggest that dairy products reduce the risk of MASLD. Evidence is growing that dairy products provide health benefits beyond their individual components, likely because of their complex matrices [11,17]. Compared with the HFD control, both MILK and YOG had lower hepatic lipid droplet formation and depleted lipid classes involved in lipid storage, such as TGs and DGs. DGs are biomarkers of lipid-induced IR, playing a role in lipid droplet formation and contributing to increased lipotoxicity [35,36] because, as intermediates in the DNL pathway, they accumulate in the liver and impair insulin signaling by activating protein kinase C-ε. In contrast, enrichment of lipid species associated with membrane curvature and fluidity, including specific PE species, was observed. The presence of certain PE species, such as PE 36:4, PE 18:1_22:6, PE 16:2_20:4, and PE 20:4_22:6, in response to MILK and YOG supplementation may modulate membrane dynamics to improve hepatic insulin signaling and fatty acid metabolism due to their roles in maintaining membrane fluidity and curvature [37]. This observation aligns with the activation of the AMPK pathway observed in the YOG and MILK groups. The pAMPK cascade inactivates ACC, a key regulatory enzyme in fatty acid synthesis. The concurrent decrease in lipogenic lipid species and AMPK activation indicates coordinated suppression of DNL, contributing to reduced hepatic lipid accumulation. Similar to our findings, others report a preventive effect of yogurt on HFD-induced MASLD through the suppression of hepatic fatty acid production, with activation of AMPK leading to reduced gene expression of *Srebp1*, *Acaca*, and *Fas* in golden hamsters [38] and inhibiting lipogenic enzyme gene expression, including *Fas* and *Acaca* in mice [39].

MILK and YOG may also alleviate hepatic steatosis by activating fatty acid β-oxidation. Induction of PGC-1α, a downstream target of AMPK, increases β-oxidation [40]. Greater abundance of certain hepatic lipids, reflecting enrichment of medium-chain fatty acid metabolism, supports enhanced mitochondrial β-oxidation in the MILK and YOG groups. Similarly, elevation of specific phospholipid species, such as PC and LPC, would facilitate fatty acid oxidation and transportation into mitochondria. These shifts in hepatic lipid content are paralleled by an increase in CPT-1α (in YOG) and other regulatory transcription factors, including SIRT1 and PPAR-α (in MILK). Various milk-derived bioactive molecules induce AMPK activation, which links with increased abundance of SIRT1 [41,42,43], and this pathway protects against HFD-induced hepatic lipotoxicity [44]. Our results are consistent with a human tissue experiment demonstrating enhanced SIRT1 activity and expression in adipocytes and myocytes incubated with serum from participants consuming a high-dairy diet [45].

YOG and MILK improved overall lipid handling by reducing the surge of circulating TG that typically follows consuming an HFD. Moreover, YOG decreased hepatic sphingolipids, long-chain fatty acids, and ceramides, which are all associated with inflammation and hepatic fat accumulation. Therefore, their decrease signifies reduced inflammation (although we did not directly measure inflammatory outcomes) and a healthier metabolism [46,47]. Some studies investigating the matrix effects of dairy products on fasting TG found a neutral effect [21,48], but data on non-fasting TG are scarce in human and rodent studies. The postprandial TG-lowering ability of yogurt and milk tested in human intervention trials [49,50] supports their ability to attenuate harmful lipid spikes following HFD intake. Several mechanisms that uniquely contribute to high TG in the fed state have been proposed, many of which are tied to IR [51].

A balanced gut microbiota community is crucial for sustaining metabolic homeostasis in the body. Dysbiosis of the microbiome is directly and indirectly associated with many metabolic disorders, including obesity, MASLD, and metabolic syndrome [52,53]. The health benefits of yogurt are attributed to rebalancing the microbiota community because of its probiotic content [54]. Yogurt starter culture organisms like *S. thermophilus* produce metabolites during milk fermentation, including branched-chain hydroxy acids [55], which improve insulin action and suppress hepatic glucose production [20]. Our data demonstrated *Streptococcus* enrichment in the gut microbiome of the YOG group, consistent with the results of others in both human [56] and animal studies [57,58]. These bacteria may ameliorate hepatosteatosis by increasing the production of short-chain fatty acids (SCFAs) and other metabolites that modulate lipid synthesis and storage pathways [59]. The YOG group also had depleted *Lachnospiraceae* and *Evtepiain* in fecal samples, both of which, at higher abundance, contribute to gut dysbiosis [60]. By modulating the gut microbiota, yogurt can mitigate the adverse effects of an HFD by leading to a healthier gut environment, improving insulin sensitivity, and reducing the risk of hepatic steatosis.

MILK increased beneficial gut bacteria such as *Anaerotignum_189125* and *Oscillospiraceae_88309* and reduced potentially harmful bacteria from the genera *Avispirillum*, *Longicatena*, and *Turicibacter*, indicative of a healthier gut environment [61]. Reduced abundance of *Anaerotignum* and *Oscillospiraceae* genera increases gut permeability, enhancing the translocation of endotoxins like lipopolysaccharides into circulation, leading to inflammation and IR [62]. These genera produce the SCFA butyrate, which modulates hepatic glucose and lipid metabolism by an AMPK-dependent mechanism [63], consistent with our liver metabolism findings. Increased milk consumption in humans has been associated with an elevation in the population of short-chain fatty acid (SCFA)-producing bacteria, including genera such as *Bifidobacterium* and *Lactobacillus*, as well as members of the Akkermansia, Lachnospiraceae, and Blautia families. Additionally, milk consumption has been shown to decrease the presence of potentially harmful bacteria, including Proteobacteria, Erysipelotrichaceae, Desulfovibrionaceae, and Clostridium perfringens [64,65].

The increase in *Sporofaciens* and *Streptococcus* in the CHE group is consistent with enhanced gut health and a lower propensity to develop systemic inflammation; however, these changes did not lead to improvement in IR or hepatic TG accumulation. Enrichment of *Streptococcus* was expected because it is used in the cheese-making process [58,59]. The CHE group also exhibited a depletion of *Limivicinus*; however, the health-related characteristics of this genus are not well determined.

Compared with MILK and YOG interventions, the response pattern to CHE is quite different, consistent with its distinct nutrient profile and matrix. CHE attenuates body weight gain and improves the lean-to-fat ratio compared with HFD, to a lesser extent than MILK and YOG, as discussed elsewhere [22]. However, the impact of cheese on insulin sensitivity and liver metabolism is less pronounced. Regarding fasted hepatic lipid metabolism, increased phosphorylation of ACC may indicate suppression of DNL, consistent with a shift toward reduced hepatic lipid synthesis and the smaller lipid droplets compared with HFD. Moreover, only minor changes were observed in the lipidomic analysis for CHE compared with HFD. This indicates that although cheese supplementation improves glucose and lipid metabolism, the connection with changes in hepatic lipid species is not as pronounced as was observed with MILK and YOG supplementation. Hanning et al. report that both low- and regular-fat cheese enhance insulin sensitivity and reduce hepatic glucose production in obese rats by suppressing hepatic PEPCK (similar to our finding), with no profound effect on fat accumulation in the liver [66]. As such, the relationship between cheese consumption and metabolic health is complex and may be context-dependent, thus warranting further research.

Emerging evidence suggests the importance of microRNAs in metabolic and digestive health [29,67]. Dairy product consumption can impact metabolism through two primary microRNA-related mechanisms. First, exosomes found in milk are rich in bioactive microRNAs, which can be absorbed intact into circulation because of their micelle-like structure. These exosomal microRNAs modulate the expression of genes regulating lipid metabolism, potentially affecting hepatic lipid accumulation and insulin sensitivity [68]. Second, dietary interventions can alter the expression of hepatic microRNAs, leading to changes in the gene expression of enzymes involved in lipid metabolism and the pathogenesis of MASLD [69]. Thus, future studies should explore how dairy consumption affects both the intake of exogenous microRNAs and the modulation of endogenous microRNA expression.

## 4. Materials and Methods

### 4.1. Experimental Procedures

All procedures involving animals were approved by the University of Alberta’s Animal Care and Use Committee (AUP00003066), following guidelines issued by the Canadian Council on Animal Care. Reporting follows the ARRIVE guidelines [70]. Eighty male C57BL/6 mice were obtained at 6 weeks of age from Charles River (St. Constant, QC, Canada) and were maintained at 23 ± 1 °C in a humidity-controlled room (50 ± 10%) under an alternating 12:12 light-dark cycle (lights on at 10:00 p.m.). Following a week of acclimatization and access to ad libitum standard chow and water, the mice were weighed and randomly assigned by cage into either a low-fat diet (LFD; n = 16) consisting of 10 kcal% fat from soybean oil and lard with a digestible energy of 3.82 kcal/g (D12450H Research Diets, New Brunswick, NJ, USA) or a HFD (n = 64) consisting of 45% of kcal from soybean oil and lard with a digestible energy of 4.73 kcal/g (D12451 Research Diets) for 7 days. Both diets had equal amounts of sucrose and protein. While mice in the LFD group remained on the same diet throughout the study, mice in the HFD groups were re-randomized into 1 of 4 groups (n = 16 per group, in 4 cages) as shown in Figure 6: (1) HFD, (2) HFD + 3.0 mL of fat-free milk (MILK; Dairyland, Saputo Dairy Products, Montreal, Canada), (3) HFD + 2.1 mL of fat-free plain yogurt (YOG; ASTRO^®^ Original Balkan Plain, Lactalis Group, Toronto, Ontario, Canada), and (4) HFD + 360 mg of reduced-fat (19% fat) cheddar cheese (CHE; Armstrong Old Light Cheddar Cheese, Saputo Dairy Products). Mice were followed for 56 days (8 weeks), with the provision of dairy products on 5 out of 7 days each week. Body weight and food intake measurements were recorded weekly, and dietary and energy intake were calculated. The study design and procedures are described in greater detail in our previous publication [22], which also describes energy metabolism outcomes and their association with the brown adipose tissue phenotype in these mice.

### 4.2. Energy Expenditure and Blood Collection

The measurements of energy expenditure, body composition, and euthanasia procedures of these mice were described in detail in our previous publication [22]. Briefly, indirect calorimetry (using Oxymax (CLAMS), Columbus Instruments, Columbus, OH, USA) and body composition analysis (using EchoMRI, Echo Medical Systems LLC, Houston, TX, USA) were performed during days 50 to 55 of the intervention. Blood glucose was measured from the tail vein using a glucometer (Contour^®^Next, Mississauga, ON, Canada) at D0 and D55 after 12 h of fasting. At the end of week 8, to evaluate metabolic markers in the fasting and refed states, all mice were subjected to a 12 h fast or a 12 h fast followed by a 4 h refeeding period with their corresponding background diet before euthanasia with CO_2_. Blood was collected and centrifuged (5000× *g*, 4 °C, 30 min), and the serum was stored at −80 °C until analysis. The entire liver was removed and weighed, and then samples were divided into fixative for histopathology analysis or frozen in liquid nitrogen and stored at −80 °C for enzyme and metabolite analyses.

### 4.3. Biochemical Analyses of Liver and Serum

Lipids were extracted from thawed liver tissue using a modified version of the Folch method [71], as described previously [72]. The dried extract was then resuspended in H_2_O and kept at −80 °C until further assessment. Serum and liver total cholesterol (TC) and TG content were measured spectrophotometrically using commercial kits (InfinityTM, Thermo Scientific, Waltham, MA, USA). Liver TG and TC content were normalized to the total protein content of the liver, which was quantified using the Pierce™ Bicinchoninic acid Protein Assay Kit (Thermo Fisher Scientific, Rockford, IL, USA).

The serum insulin concentration of 12 h fasted mice was measured using a mouse insulin ELISA kit (ALPCO, Salem, NH, USA). IR was estimated with the homeostatic model assessment-insulin resistance (HOMA-IR) formula: HOMA-IR = [fasting plasma insulin (µU/mL) × fasting blood glucose (mmol/L)/22.5].

### 4.4. Histological Staining

The liver samples for histology were fixed in 10% formalin, embedded in paraffin, and cut into 5 µm sections. Following deparaffinization, tissue sections were stained with hematoxylin and eosin (H&E). For each mouse, a minimum of 12 light microscopic images was taken by one researcher at 20X magnification (Axio Observer A1 with AxioCam HRc, Germany). The lipid droplet area of each image was manually quantified using ImageJ (Version 1.54g, National Institutes of Health, Baltimore, MD, USA) [73]. The researcher was blinded during image capture and quantification. Based on the quantification data, representative images were selected for presentation.

### 4.5. Immunoblot Analysis

To investigate hepatic enzyme abundance and phosphorylation, 50 mg of frozen liver tissue was homogenized in RIPA lysis buffer, centrifuged, and the supernatant collected. Total protein concentrations were determined by the Pierce^TM^ bicinchoninic acid method. Samples (2 μg/μL) were prepared in Laemmli protein sample buffer (4X; Bio-Rad, Hercules, CA, USA) in the presence of 2-mercaptoethanol. Sample proteins were separated using 8% to 12% sodium dodecyl sulfate-polyacrylamide gel electrophoresis and transferred to nitrocellulose membranes. Membranes were blocked with 5% skim milk or 5% bovine serum albumin (BSA) in phosphate-buffered saline-0.05% Tween for 1 h. Subsequently, membranes were incubated with primary antibodies (Appendix A) at a dilution of 1:1000 or 1:500, based on the manufacturer’s recommendation, in 2.5% BSA overnight at 4 °C. Membranes were then incubated with horseradish peroxidase-conjugated secondary antibodies (Appendix A) at a dilution of 1:5000 in 2.5% BSA at room temperature for 1 h. Blots were incubated with enhanced chemiluminescence (ThermoFisher Scientific, Rockford, IL, USA) for 1 min, and the protein bands were imaged using a ChemiDoc MP Imager (Bio-Rad Laboratories, CA, USA) and analyzed using Image Lab software (Bio-Rad version 6.1.0).

### 4.6. Lipidomics

The Metabolomics Innovation Centre (University of Alberta, Canada) conducted a comprehensive analysis of liver lipid extracts from 4 randomly selected samples per treatment group, each pooled from 2 mice. Untargeted LC-MS and LC-MS/MS methods were used for global lipidomics analysis [74,75]. A detailed description of the procedures is provided elsewhere [22]. Identified features were normalized using internal standards and the median intensity ratio. Statistical analysis was performed in MetaboAnalyst 6.0 (https://www.metaboanalyst.ca/ (accessed on 15 May 2024)). The pathway enrichment analysis was conducted in the Lipid Ontology (LION) enrichment analysis web application (LION/web, www.lipidontology.com (accessed on 1 June 2024)) by inputting the compound details of altered lipid species (*p*-values < 0.05) for each dairy product group compared with the HFD group.

### 4.7. Gut Microbiota Analysis

Fecal samples were collected under sterile conditions on the morning of day 49 (week 7) and immediately frozen at −80 °C until 16S rRNA amplicon gene sequencing was performed. For every 100 mg feces sample, 700 µL of lysis solution (Zymo Research Corp, Irvine, CA, USA) and 100 µL of Roth zirconia/silica beads (0.1 mm in diameter) were added. Bacteria were lysed by mechanical disruption 3 times for 5 min each, with 5 min on ice in between, using a Mini-Bead Beater-96 (BioSpec, Bartlesville, OK, USA). Subsequently, the homogenate was centrifuged, and 200 µL of supernatant was transferred into a 96-well deep-well plate (Nunc, ThermoFisher Scientific, Rochester, NY, USA) for purification to remove debris, particulate matter, and beads. DNA was extracted using 40 µL of beads on a Tecan Fluent following the manufacturer’s instructions (Zymo Research Corp, Irvine, USA). The 16S rRNA gene amplification of the V4 region (forward: CCTACGGGNGGCWGCAG, reverse: GACTACHVGGGTATCTAATCC) was carried out according to an established protocol [76,77]. DNA was standardized to 25 ng/μL and used for sequencing polymerase chain reaction (PCR), including unique 12-base Golay barcodes with specific primers (Sigma-Aldrich, Saint Louis, MI, USA). For every sample, PCR was carried out in triplicate using Q5 polymerase (New England Biolabs, Ipswich, MA, USA) under the following conditions: 30 s initial denaturation at 98 °C, followed by 25 cycles of 10 s at 98 °C, 20 s at 55 °C, and 20 s at 72 °C. PCR amplicons were sequenced using 300 bp paired-end sequencing (PE300) on an Illumina MiSeq Sequencing Platform 300PE after pooling and standardization to 10 nmol/L.

The resulting raw reads were demultiplexed by idmp (https://github.com/yhwu/idemp (accessed on 8 June 2024)) according to the provided barcodes [77]. The libraries underwent processing that included merging the paired-end reads, filtering low-quality sequences, dereplication to identify unique sequences, singleton removal, denoising, and chimera checking using the USEARCH pipeline version 11.0.667 [78]. Reads were merged using the fastq_mergepairs command (parameters: maxdiffs 30, pctid 70, minmergelen 200, maxmergelen 400) and filtered for low quality with fastq_filter (maxee 1), and singletons were removed using the fastx_uniques command (minuniquesize 2).

To predict biological sequences (ASV, zOTUs) and filter chimeras, the unoise3 command (minsize 10, unoise_alpha 2) was used, following the amplicon quantification with the usearch_global command (strand plus, id 0.97, maxaccepts 10, top_hit_only, maxrejects 250). Taxonomic assignment was performed by Constax (classifiers: rdp, sintax, blast) using the SILVA138 database [79,80], summarizing into a biom file for visualization and downstream analysis.

The processed data were then uploaded into Microbiome Analyst [81] using a low count filter with a prevalence threshold of 10% and a low variance filter with a threshold of 10%. The number of observed OTUs, Shannon index, and Simpson’s reciprocal indices were used to assess α-diversity. The Bray–Curtis dissimilarity, weighted UniFrac distances, and unweighted UniFrac distances were used to measure β-diversity. Principal coordinate analysis (PCoA) was used for data visualization, and the dissimilarities were assessed using PERMANOVA. To evaluate the comparative abundance of different taxa between groups, DESeq2 [82] was used at the genus level.

### 4.8. Statistical Analysis

Data were analyzed using GraphPad Prism 9 (GraphPad Software Inc., San Diego, CA, USA), R (4.3.3, https://www.r-project.org/ (accessed between 1 January–31 July 2024)), or SPSS 17 (IBM Corp., Armonk, NY, USA). Significance for statistical tests was set to *p* < 0.05 (two-sided) unless stated otherwise. Mean ± SEM was used to present all parameters except lipidomics and microbiome data, which were presented as relative fold changes. The Student’s *t*-test was used to compare parameters between the LFD and HFD groups. One-way ANOVA followed by Tukey’s post hoc test was used to determine differences between the MILK, YOG, CHE, and HFD groups. The weight-adjusted energy expenditure, estimated based on the residual regression model, used body weight as the independent variable and absolute energy expenditure as the dependent variable.

## 5. Conclusions

In summary, MILK and YOG supplementation in the HFD model significantly lowered hepatic steatosis, consistent with changes in the abundance of enzymes that predict reduced DNL, enhanced β-oxidation, and improved insulin signaling in the liver. MILK and YOG also improved postprandial triglyceride handling. Conversely, CHE weakly impacted insulin sensitivity and lipid metabolism despite also lowering hepatic steatosis. These effects of dairy feeding are potentially mediated through modulation of specific lipid species and gut microbiota composition. The findings suggest that dairy products, especially milk and yogurt, may help reduce obesity and related metabolic issues, such as liver steatosis, through mechanisms involving glucose and lipid metabolism, possibly mediated by modulation of the gut microbiome.

## Figures and Tables

**Figure 1 ijms-26-05026-f001:**
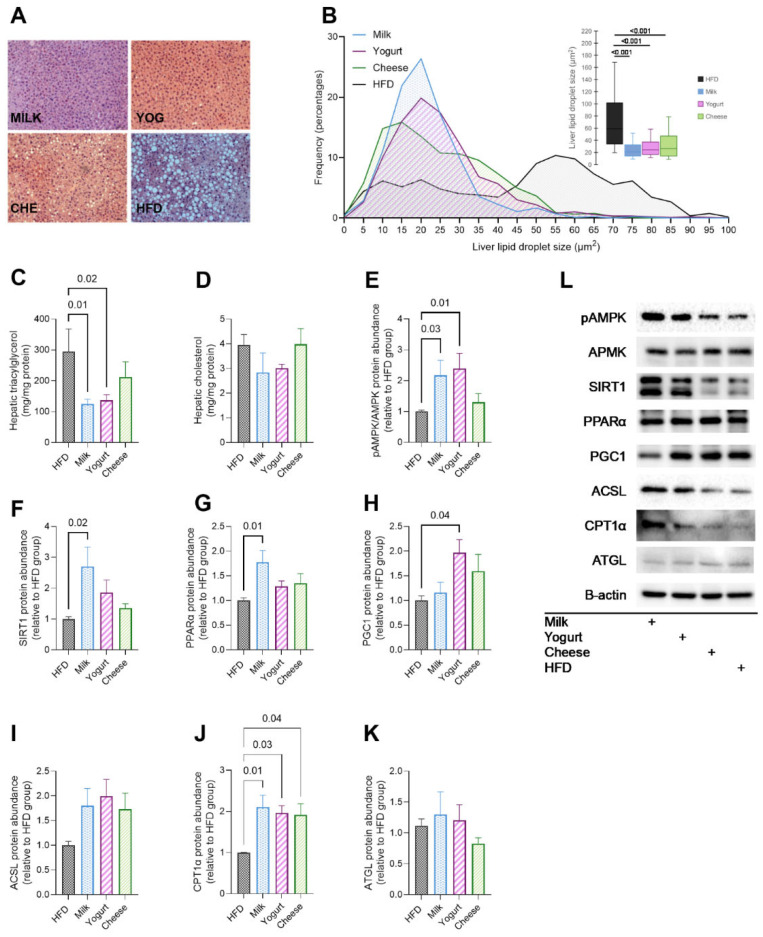
Effects of milk, yogurt, and cheese on hepatic steatosis in HFD-fed C57BL/6 mice. (**A**) Representative microscopic images of liver sections from each group stained with H&E. (**B**) Frequency distribution of the liver lipid droplet size and box plot quantification of the liver lipid droplet area. Colorimetric quantification of (**C**) triacylglycerol (TG) and (**D**) total cholesterol (TC) concentrations in the liver. Western blot results indicate liver lipid metabolism pathway enzyme abundance in the fasting state for lipid oxidation, including (**E**) pAMPK/AMPK, (**F**) SIRT1, (**G**) PPARα, (**H**) PGC1α, (**I**) ACSL, (**J**) CPT1α, and (**K**) lipolysis enzyme ATGL. (**L**) Representative immunoblots (fasted state; a loading control (B-actin) was performed for each blot). Data are presented as mean ± SE of n = 6–8 mice. *p*-values are indicated for pairwise comparisons following ANOVA and Tukey’s post hoc test.

**Figure 2 ijms-26-05026-f002:**
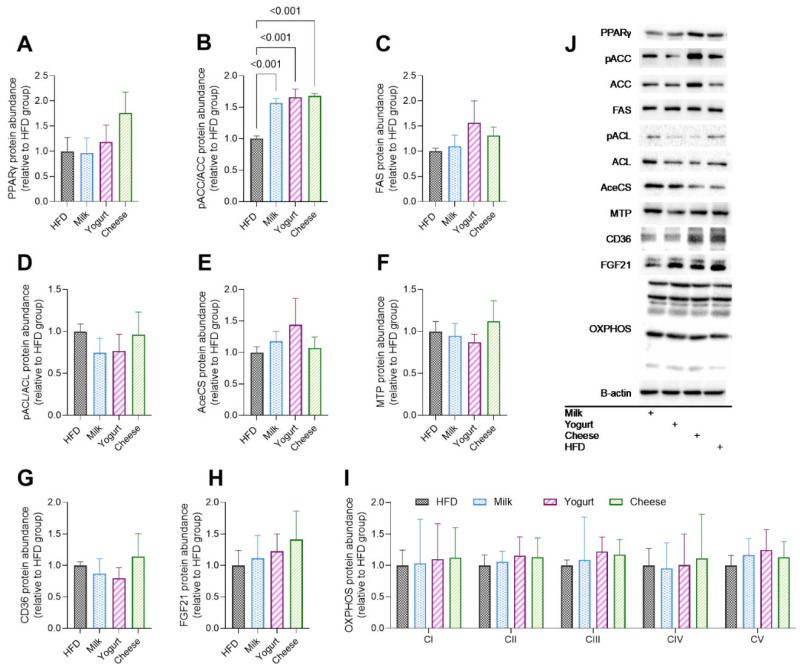
Effects of milk, yogurt, and cheese on regulatory proteins involved in hepatic de novo lipogenesis, lipid uptake, and export in HFD-fed C57BL/6 mice in the fasting state. Western blot results of enzymes involved in de novo lipogenesis, including (**A**) PPARγ, (**B**) pACC/ACC, (**C**) FAS, (**D**) pACL/ACL, and (**E**) AceCS. Relative abundance of proteins involved in lipid export (**F**) MTP and lipid uptake (**G**) CD36. Overall energy metabolism-related enzymes include (**H**) FGF21 and (**I**) OXPHOS complex subunits. (**J**) Representative immunoblots (fasted state; loading control (B-actin) was performed for each blot). Data are presented as mean ± SE of n = 6–8 mice. *p*-values are indicated for pairwise comparisons following ANOVA and Tukey’s post hoc test.

**Figure 3 ijms-26-05026-f003:**
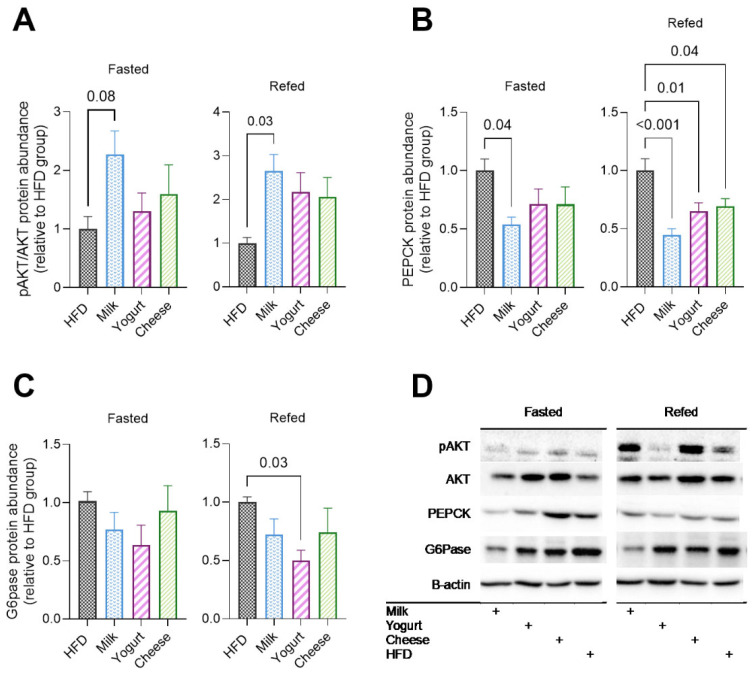
Effects of milk, yogurt, and cheese on enzymes involved in hepatic insulin signaling and gluconeogenesis in HFD-fed C57BL/6 mice in the fasted and refed states. Western blot results of a key regulator of insulin signaling, (**A**) pAKT/AKT, and rate-limiting gluconeogenesis enzymes, (**B**) PEPCK and (**C**) G6Pase. (**D**) Representative immunoblots (fasted and refed state; loading control (B-actin) was performed for each blot). Data are presented as mean ± SE for n  =  6–8 mice. *p*-values are indicated for pairwise comparisons following ANOVA and Tukey’s post hoc test.

**Figure 4 ijms-26-05026-f004:**
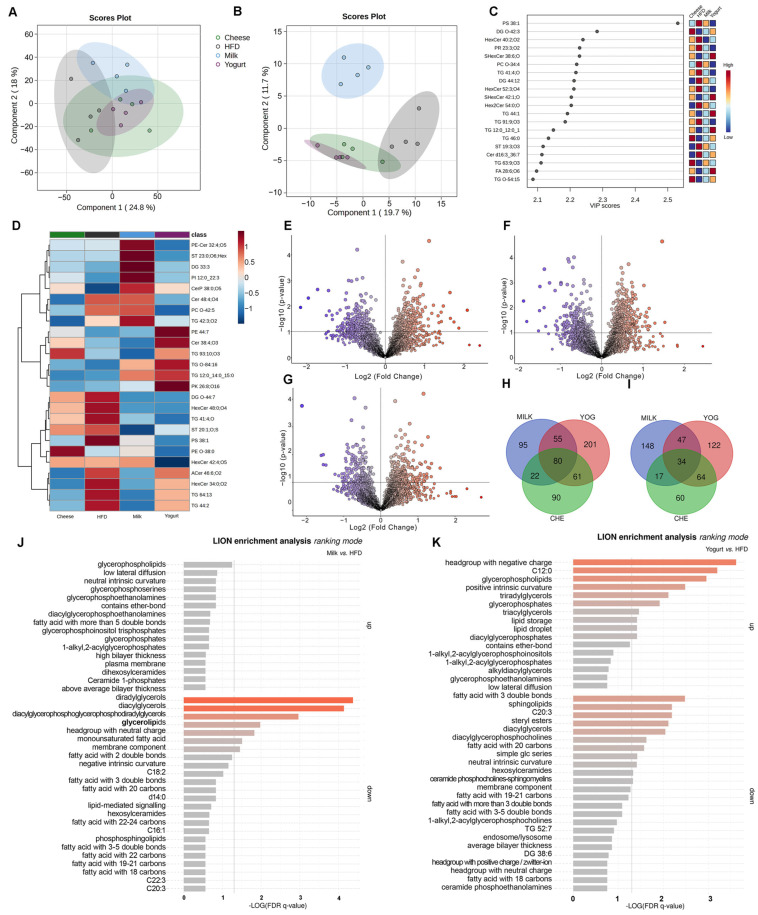
Lipidomic and LION enrichment analysis of liver samples from HFD-fed C57BL/6 mice (n = 4) treated with milk, yogurt, and cheese. (**A**) PLS-DA scores plot showing separation between groups based on liver lipidomic profiles. (**B**) sPLS-DA scores plot depicting clustering of lipidomic data by intervention groups using the 100 most significant lipid species. (**C**) VIP scores from the sPLS-DA model identify the top lipid species contributing to group separation. (**D**) Heatmap of the top 25 lipid species with the most significant differences between intervention groups. (**E**–**G**) Volcano plots displaying differential abundance of lipid species in (**E**) MILK vs. HFD, (**F**) YOG vs. HFD, and (**G**) CHE vs. HFD. (**H**,**I**) Venn diagrams illustrating the increased (**H**) and decreased (**I**) number of lipid species shared between intervention groups compared to HFD. (**J**,**K**) LION enrichment analysis revealing significantly altered lipid pathways in (**J**) MILK vs. HFD and (**K**) YOG vs. HFD.

**Figure 5 ijms-26-05026-f005:**
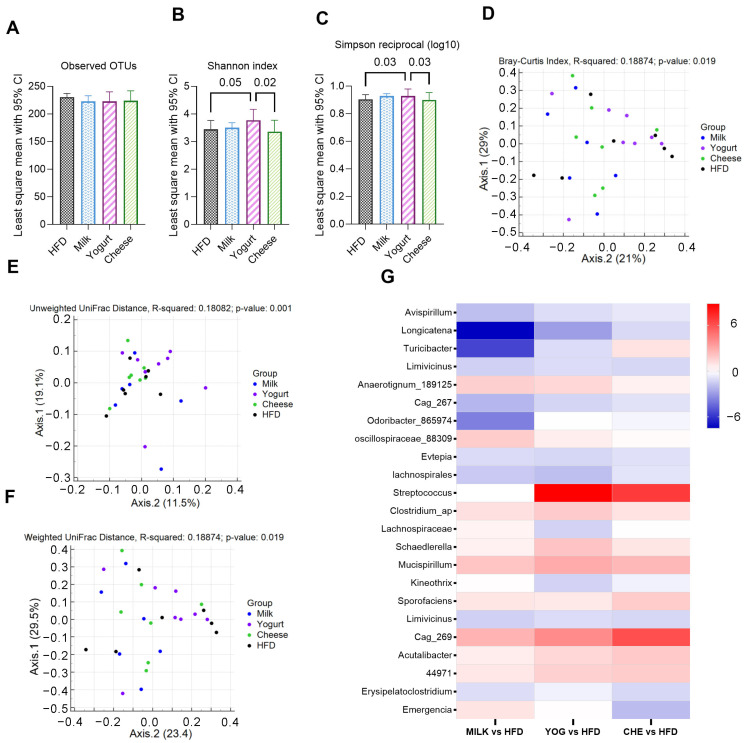
Effects of milk, yogurt, and cheese on gut microbiota composition in HFD-fed C57BL/6 mice (n = 7–8). (**A**–**C**) Alpha-diversity indices: (**A**) observed OTUs, (**B**) Shannon index, and (**C**) Simpson reciprocal (log10), indicating the least square mean with 95% confidence intervals (95% CI). Significant *p*-values (*p* < 0.05) are illustrated for pairwise comparisons. (**D**–**F**) Beta-diversity analysis: (**D**) Bray–Curtis index, (**E**) unweighted UniFrac distance, and (**F**) weighted UniFrac distance. (**G**) Heatmap displaying the relative abundance of bacterial genera that were significantly (*p* < 0.05) altered in the dairy intervention groups compared with the HFD group.

**Figure 6 ijms-26-05026-f006:**
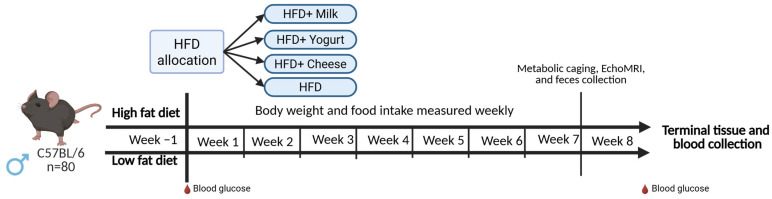
Study design and experimental timeline for assessing the effects of milk, yogurt, and cheese on metabolic health in HFD-fed C57BL/6 mice.

**Table 1 ijms-26-05026-t001:** Effects of milk, yogurt, and cheese on body weight, calorie intake, and biochemical markers ^1,2^.

	Groups (n = 16)			
	HFD	MILK	YOG	CHE
Baseline body weight (g)	25.6 (0.3) ^a^	24.6 (0.4) ^a^	24.8 (0.4) ^a^	25.1 (0.3) ^a^
Final body weight (g)	39.5 (0.9) ^a^	33.4 (0.3) ^b^	33.5 (0.8) ^b^	35.4 (1.0) ^b^
Body weight gain (g)	13.9 (0.9) ^a^	8.5 (0.9) ^b^	9.2 (0.7) ^b^	10.3 (1.2) ^b^
Finalfat body mass (g)	12.6 (0.5) ^a^	8.5 (0.6) ^b^	8.7 (0.6) ^b^	9.9 (0.7) ^b^
Final lean body mass (g)	21.9 (1.1) ^a^	20.6 (0.7) ^b^	20.8 (1.2) ^b^	21.2 (0.9) ^a,b^
Lean-to-fat mass ratio	1.7 (0.3) ^a^	2.7 (1.1) ^b^	2.6 (0.7) ^b^	2.3 (0.8) ^a,b^
Average energy intake (kcal/day/mouse) ^3^	12.6 (0.2) ^a^	12.0 (0.1) ^a^	12.3 (0.1) ^a^	11.9 (0.3) ^a^
Energy expenditure (kcal/d) ^4^	11.6 (0.2) ^a^	13.1 (0.2) ^b^	12.8 (0.2) ^b^	12.9(0.2) ^b^
Baseline fasting blood glucose (mmol/L)	8.1 (0.4) ^a^	8.2 (0.3) ^a^	8.0 (0.4) ^a^	7.5 (0.3) ^a^
Final fasting blood glucose (mmol/L)	10.6 (0.3) ^a^	9.3 (0.3) ^b^	9.2 (0.2) ^b^	9.4 (0.4) ^b^
Final serum insulin (pmol/L; n = 8)	158.5 (31.5) ^a^	61.9 (6.9) ^b^	94.7 (18.8) ^a,b^	94.7 (17.0) ^a,b^
HOMA-IR (n = 8)	3.4 (0.7) ^a^	1.3 (0.2) ^b^	2.0 (0.3) ^a,b^	2.0 (0.4) ^a,b^
Final fasting serum triacylglycerol (mg/dL; n = 8) ^5^	169.2 (16.3) ^a^	157.4 (23.8) ^a^	144.4 (16.3) ^a^	149.6 (11.6) ^a^
Final refed serum triacylglycerol (mg/dL; n = 8) ^6^	308.5 (41.9) ^a^	191.1 (14.3) ^b^	208.6 (16.6) ^b^	240 (19.6) ^a,b^
Final fasting serum cholesterol (mg/dL; n = 8)	23.2 (3.3) ^a^	23.9 (1.3) ^a^	22.9 (1.2) ^a^	23.1 (1.8) ^a^
Final refed serum cholesterol (mg/dL; n = 8)	23.1 (2.7) ^a^	20.4 (1.6) ^a^	25.5 (1.8) ^a^	20.4 (1.2) ^a^

^1^ Values are expressed as mean (standard error of the mean). ^2^ Different superscript letters within a row indicate a significant difference (*p* < 0.05) between groups according to Tukey’s post hoc test following ANOVA. ^3^ The pellet intake (g) per cage (four mice per cage) was measured weekly, and the average daily caloric intake for each mouse was estimated using the energy density of each diet. ^4^ Energy expenditure was adjusted for body weight using the residual model. ^5^ Measured in 12 h fasted serum collected at euthanasia. ^6^ Measured in serum collected at euthanasia after refeeding for 4 h, following 12 h of fasting.

## Data Availability

The data for this study have been deposited in the European Nucleotide Archive (ENA) at EMBL-EBI under accession number PRJEB86069. Further associated data and additional results from this study are available from the corresponding author upon request.

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
