# Peer review of "A Comparison of the Effects of Milk, Yogurt, and Cheese on Insulin Sensitivity, Hepatic Steatosis, and Gut Microbiota in Diet-Induced Obese Male Mice"

_ijms, 2025, doi:10.3390/ijms26115026_

Round 1

Reviewer 1 Report

Comments and Suggestions for Authors

The article "A Comparison of the Effects of Milk, Yogurt, and Cheese on Insulin Sensitivity, Hepatic Steatosis, and Gut Microbiota in Diet-Induced Obese Male Mice" provides new insights into insulin sensitivity, hepatic steatosis, and gut microbiota in male mice. Suggestions:

  1. The Introduction lacks high-quality, relevant information related to the topic. It should include more details specifically about the gut microbiota.
  2. Section 2 should be clearly labeled as Materials and Methods.
  3. The Results are presented in a manner that lacks professionalism. The section begins directly with the mention of tables and figures, which should be integrated more smoothly into the narrative.
  4. The statistical analysis is quite basic. The results section should be significantly improved, especially considering the relatively small sample size.
  5. The Discussion section includes some extrapolations to humans. Please address two key points:
  • The role of the microbiota in human metabolic syndrome – highly recommended - https://doi.org/10.3390/jcm14062054
  • The involvement of microRNAs in digestive diseases. This could even be developed into a separate subsection under Future Perspectives, as this is a highly relevant and emerging topic – highly recommended - https://doi.org/10.3390/jcm14082678
  1. The Conclusions should be shortened.

Author Response

  1. The Introduction lacks high-quality, relevant information related to the topic. It should include more details, specifically about the gut microbiota.

Response: We appreciate the reviewer for highlighting this point. In the revised manuscript, we have substantially edited the introduction to incorporate more recent publications and to expand upon the rationale concerning the impact of the gut microbiome on the development of metabolic diseases. Additionally, we elaborate on the significance of rebalancing the gut microbiome through the consumption of fermented foods, which may play a role in preventing metabolic diseases, including MASLD.

The revised version is highlighted on page 2, lines 44-48 and 62-79.

  1. Section 2 should be clearly labeled as Materials and Methods.

Response: The second part of the manuscript has been clearly labeled as Materials and Methods.

  1. The Results are presented in a manner that lacks professionalism. The section begins directly with the mention of tables and figures, which should be integrated more smoothly into the narrative.

Response: The result section has been thoroughly revised, and the text has been referenced to figures and tables and integrated into the narrative.

  1. The statistical analysis is quite basic. The results section should be significantly improved, especially considering the relatively small sample size.

Response: The sample size is consistent with ethical standards in animal research, which ask researchers to consider how to use only the number of animals required. As reported in our previous publication, a power calculation was done to estimate the required number of animals for our primary outcomes. The analysis of the in vivo phenotype variables, western blot, and related variables is consistent with what we have done in previous, related publications and consistent with standard methods of data presentation. The lipidomics analysis may be considered exploratory, given that we needed to pool samples in order to obtain sufficient volume for the process. However, we followed general statistical procedures for this type of work available in the Metaboanalyst platform, including data normalization, autoscaling, and multivariate techniques such as PCA and PLS-DA. We further complemented this with pathway-level insights using LION (Lipid Ontology) enrichment analysis, which enabled us to interpret shifts in biologically relevant lipid categories, such as those associated with membrane curvature, storage lipids, and signaling lipids, thereby enhancing the mechanistic depth of our findings. We note that Reviewer 2 considered the analysis appropriate. Regarding the microbiome data, analysis was conducted at the genus level as indicated as an appropriate level for the 16s rRNA result and because this level of taxonomic resolution offers a balanced trade-off between interpretability and statistical robustness, given our moderate sample sizes. We performed both diversity metrics and differential abundance testing using MicrobiomeAnalyst. To address the reviewer’s suggestion, we have revised the Results section to improve clarity and highlight key statistical comparisons.

  1. The Discussion section includes some extrapolations to humans. Please address two key points:
    1. The role of the microbiota in human metabolic syndrome – highly recommended - https://doi.org/10.3390/jcm14062054
    2. The involvement of microRNAs in digestive diseases. This could even be developed into a separate subsection under Future Perspectives, as this is a highly relevant and emerging topic – highly recommended - https://doi.org/10.3390/jcm14082678

Response: We appreciate the reviewer for pointing out the role of microbiota in human metabolic syndrome and the role of microRNAs in digestive diseases and their potential modulation through dietary interventions. The association between the gut microbiome and metabolic syndrome has been integrated into the discussion when we addressed the impact of microbiome shifts in response to milk and yogurt. Furthermore, we have added a paragraph at the end of the discussion section to indicate the dual role of miRNAs in the context of dairy consumption and hepatic lipid metabolism.

The revised version is highlighted on page 13, lines 416 - 426.

  1. The Conclusions should be shortened.

Response: The conclusion has been revised to enhance precision by concentrating on the primary finding of the study.

Reviewer 2 Report

Comments and Suggestions for Authors
  1. The lipidomic analysis  is extensive, identifying over 400 differentially abundant species. The use of multivariate methods are appropriate, but the discussion tends to focus on group separation rather than the biological significance of specific lipid classes. For instance, several species of phosphatidylethanolamines and diacylglycerols appear to differ between groups, but their functional implications are not addressed. Including a more focused interpretation of these lipid shifts especially in relation to hepatic insulin signaling or steatosis would improve the mechanistic narrative.
  2. The LION enrichment analysis is a helpful addition, though its outputs are not directly explained in the main text. If certain lipid ontology categories (storage lipids, membrane curvature, or signaling lipids) are enriched or depleted in response to treatment, it would be helpful to briefly interpret how these categories relate to hepatic lipid metabolism or insulin responsiveness.

  3. One limitation worth noting is that insulin sensitivity is inferred primarily from fasting indices (HOMA-IR) and post-refeeding hepatic markers such as pAKT. While these are informative, the absence of dynamic tests ( GTT or ITT) should be acknowledged, and the interpretation of insulin sensitivity results should be framed accordingly. This does not detract from the value of the data, but a more cautious tone may be warranted when discussing improvements in insulin action.

  4. In the current form, the discussion presents the findings in separate domains metabolic, microbial, and lipidomic without attempting to integrate them. Given the strength of the dataset, a more synthetic view would be welcome. For example, are the changes in Streptococcus or Anaerotignum plausibly linked to shifts in specific hepatic lipid species or metabolic markers? Even speculative connections, clearly labeled as such, can help contextualize the results.

Overall, this is a well-conducted study that makes a meaningful contribution to understanding how different dairy products may affect metabolic health. With additional interpretation in key areas and some attention to mechanistic coherence, the manuscript could be further improved and positioned as a useful reference for future research.

Author Response

Reviewer #2

  1. The lipidomic analysis is extensive, identifying over 400 differentially abundant species. The use of multivariate methods is appropriate, but the discussion tends to focus on group separation rather than the biological significance of specific lipid classes. For instance, several species of phosphatidylethanolamines and diacylglycerols appear to differ between groups, but their functional implications are not addressed. Including a more focused interpretation of these lipid shifts, especially in relation to hepatic insulin signaling or steatosis, would improve the mechanistic narrative.

Response: We appreciate the reviewer’s comment. The discussion section has been thoroughly revised to address all reviewer comments and enhance the narrative flow. To address the reviewer's concern, we have further elaborated on the mechanistic insights of lipid shifts, rather than solely on group separation. We have specifically explained in greater detail the functions of PEs and DGs and their impact on insulin resistance and fat accumulation in the liver. Additionally, relevant references have also been included to support these interpretations.​

The revised version is highlighted on page 2, lines 316-330, 349-352.

  1. The LION enrichment analysis is a helpful addition, though its outputs are not directly explained in the main text. If certain lipid ontology categories (storage lipids, membrane curvature, or signaling lipids) are enriched or depleted in response to treatment, it would be helpful to briefly interpret how these categories relate to hepatic lipid metabolism or insulin responsiveness.

Response: In the current revised version of our manuscript, we have integrated the enriched and depleted lipid ontology terms identified in the LION analysis and discussed their significance to hepatic lipid metabolism and insulin responsiveness in the discussion section. 

The revised version is highlighted on pages 11-12, lines 323-337 and 342-354.

  1. One limitation worth noting is that insulin sensitivity is inferred primarily from fasting indices (HOMA-IR) and post-refeeding hepatic markers such as pAKT. While these are informative, the absence of dynamic tests ( GTT or ITT) should be acknowledged, and the interpretation of insulin sensitivity results should be framed accordingly. This does not detract from the value of the data, but a more cautious tone may be warranted when discussing improvements in insulin action.

Response: We acknowledge the limitation regarding the lack of dynamic assessment of insulin sensitivity, and the interpretations are drawn based on surrogate markers, namely HOMA-IR and hepatic pAKT levels, which, while informative, do not capture the dynamic aspects of insulin responsiveness. We have addressed the limitation of lacking the dynamic test of glucose homeostasis in the discussion section as follows:

The revised version is highlighted on page 11, lines 317-319.

  1. In the current form, the discussion presents the findings in separate domains metabolic, microbial, and lipidomic without attempting to integrate them. Given the strength of the dataset, a more synthetic view would be welcome. For example, are the changes in Streptococcus or Anaerotignum plausibly linked to shifts in specific hepatic lipid species or metabolic markers? Even speculative connections, clearly labeled as such, can help contextualize the results.

Response: We appreciate the reviewer’s suggestion. We believe that addressing the reviewer’s comment enhances the understanding of the metabolic effects of dairy products by integrating the metabolic, microbial, and lipidomic findings to provide a more cohesive interpretation of our results. By integrating these observations, we suggest that the MILK and YOG interventions modulate the gut microbiota composition or preserve the microbiome from HFD-induced dysbiosis. This might be associated with higher production of beneficial microbial metabolites like SCFA, which subsequently impacts hepatic lipid metabolism and insulin responsiveness. We have incorporated this integrated perspective into the revised discussion section to provide a more comprehensive understanding of the interplay between diet, gut microbiota, and hepatic metabolism.

The revised version is highlighted on page 12, lines 367-377.

  1. Overall, this is a well-conducted study that makes a meaningful contribution to understanding how different dairy products may affect metabolic health. With additional interpretation in key areas and some attention to mechanistic coherence, the manuscript could be further improved and positioned as a useful reference for future research.

Respond: We truly appreciate the reviewer’s positive feedback on our study. It’s gratifying to know that our research is helping to enhance the understanding of how various dairy products can impact metabolic health. In light of the suggestion for further interpretation and a stronger connection between the assessments, we have made extensive revisions to the manuscript.

Round 2

Reviewer 1 Report

Comments and Suggestions for Authors

The authors made considerable changes to the manuscript and significantly increased its quality.